EMBO
Molecular Medicine

# Mutations in *TIMM50* compromise cell survival in OxPhos-dependent metabolic conditions

Aurelio Reyes[1],[*] , Laura Melchionda[2], Alberto Burlina[3], Alan J Robinson[1], Daniele Ghezzi[2] &
Massimo Zeviani[1],[**]

## Abstract

TIMM50 is an essential component of the TIM23 complex, the mitochondrial inner membrane machinery that imports cytosolic proteins containing a mitochondrial targeting presequence into the mitochondrial inner compartment. Whole exome sequencing (WES) identified compound heterozygous pathogenic mutations in *TIMM50* in an infant patient with rapidly progressive, severe encephalopathy. Patient fibroblasts presented low levels of TIMM50 and other components of the TIM23 complex, lower mitochondrial membrane potential, and impaired TIM23-dependent protein import. As a consequence, steady-state levels of several components of mitochondrial respiratory chain were decreased, resulting in decreased respiration and increased ROS production. Growth of patient fibroblasts in galactose shifted energy production metabolism toward oxidative phosphorylation (OxPhos), producing an apparent improvement in most of the above features but also increased apoptosis. Complementation of patient fibroblasts with TIMM50 improved or restored these features to control levels. Moreover, *RNASEH1* and *ISCU* mutant fibroblasts only shared a few of these features with *TIMM50* mutant fibroblasts. Our results indicate that mutations in *TIMM50* cause multiple mitochondrial bioenergetic dysfunction and that functional TIMM50 is essential for cell survival in OxPhos-dependent conditions.

**Keywords**  bioenergetic dysfunction; mitochondrial import; OxPhos; TIMM50
**Subject Categories**  Genetics, Gene Therapy & Genetic Disease; Metabolism

## Introduction

The mammalian mitochondrial proteome is estimated to contain about 1,500 proteins (Meisinger *et al*, 2008), but only 13 of them are encoded by the mitochondrial DNA (mtDNA) and synthesized in the mitochondrial matrix. The remaining proteins are encoded by nuclear genes, synthesized on cytosolic ribosomes and then imported into mitochondria by specific protein import machineries located on both the outer and the inner mitochondrial membranes (OMM and IMM, respectively; Sokol *et al*, 2014).

The translocase of the outer mitochondrial membrane (TOM) complex is the common entry gate for all mitochondrial precursor proteins and, in humans, it consists of eight subunits (Fig 1A). TOMM20 and TOMM70 serve as receptors for mitochondrial precursor proteins, being the latter responsible for engaging with the multichaperone complex of Hsp70/Hsp90/TOMM34. The channel-forming subunit is TOMM40 while TOMM22 both stabilizes the complex and interacts with subunits of the translocase of the inner mitochondrial membrane (TIM) complex. Additional small subunits, TOMM5, TOMM6, and TOMM7, are involved in the assembly and stability of the TOM complex (Kato & Mihara, 2008).

Over half of all mitochondrial proteins synthesized in the cytosol as precursors contain a cleavable mitochondrial targeting sequence (MTS) at their N-terminus. After crossing the OMM, these precursors are directed to the TIM23 complex. The TIM23 complex is mainly utilized by proteins targeted to the matrix but it is also partly used for proteins directed to the IMM and the intermembrane space (IMS; Dudek *et al*, 2013; Sokol *et al*, 2014; Straub *et al*, 2016). The TIM23 core complex consists of TIMM23, TIMM17, and TIMM50. TIMM17 occurs in two flavors, TIMM17A and TIMM17B, which are ubiquitously expressed in human tissues. Along with TIMM23, they form the channel in the IMM but they are never together in the same TIM23 complex, playing TIMM17B a more active role in import than TIMM17A (Sinha *et al*, 2014). TIMM50 serves as receptor that brings the presequence proteins from the TOM complex to the channel-forming subunit TIMM23 (Sokol *et al*, 2014; Schulz *et al*, 2015). In yeasts, Tim50 not only mediates the transfer of presequence proteins from Tom40 to Tim23, but it is also responsible for maintaining the Tim23-Tim17 channel closed at resting state, therefore avoiding the dissipation of membrane potential (Meinecke *et al*, 2006). Two isoforms of TIMM50 have been described in humans: A long TIMM50L (456 amino acids) isoform targeted to the nucleus due to the presence of an internal nuclear localization signal (NLS; Xu *et al*, 2005) and a short TIMM50S (353 amino acids) isoform found in mitochondria as part of the TIM23 complex (Guo *et al*, 2004). Both isoforms have a helical transmembrane domain (TMD)

1   MRC Mitochondrial Biology Unit, University of Cambridge, Cambridge, UK
2   Unit of Molecular Neurogenetics, Foundation Carlo Besta Neurological Institute-IRCCS, Milan, Italy
3   Division of Inherited Metabolic Diseases, Department of Pediatrics, University Hospital Padova, Padova, Italy
    *Corresponding author. Tel: +44 1223 252843; E-mail: art@mrc-mbu.cam.ac.uk
    **Corresponding author. Tel: +44 1223 252702; Fax: +44 1223 252705; E-mail: mdz21@mrc-mbu.cam.ac.uk

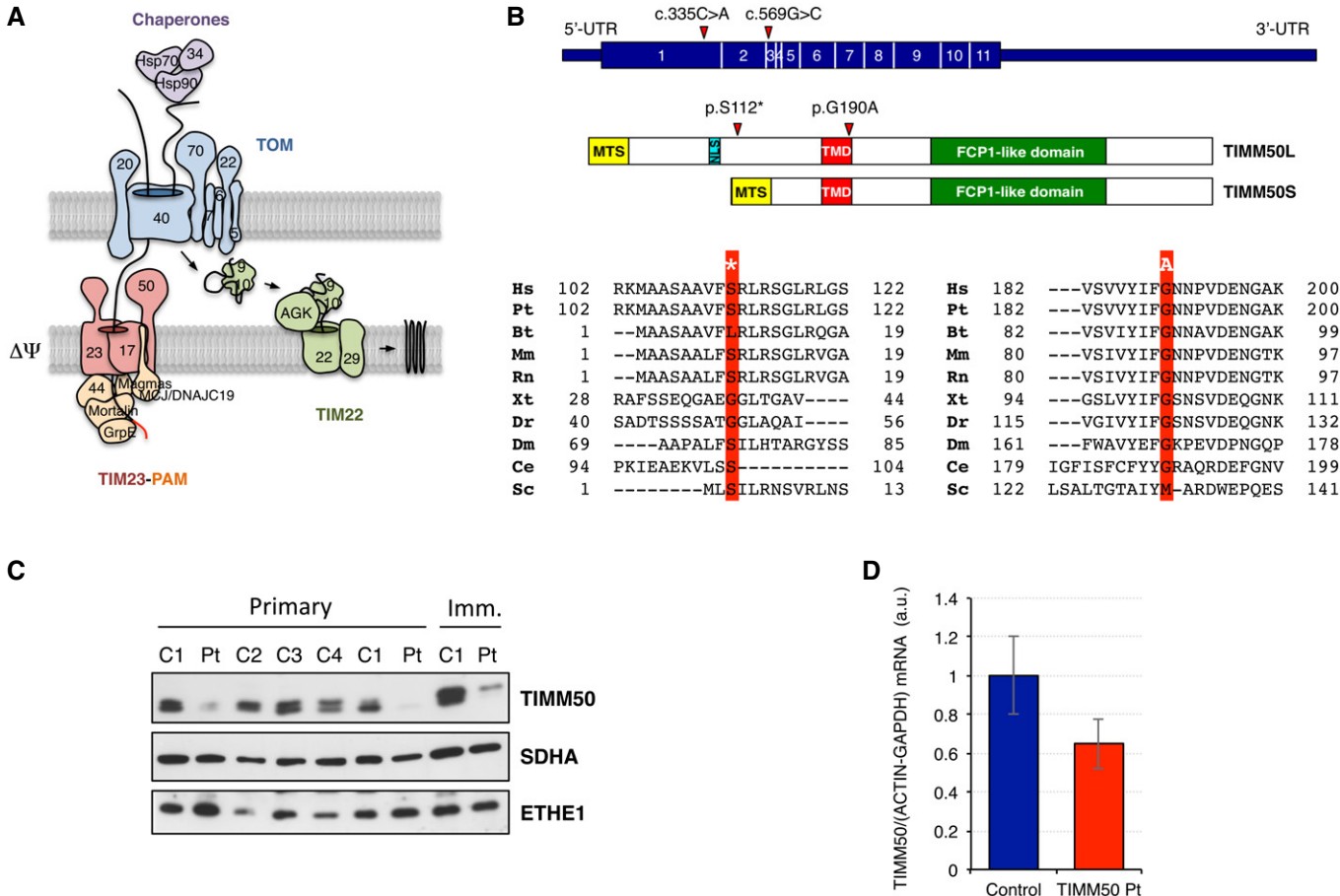

**Figure 1. TIMM50 and the import machinery.**

A  Schematic structure of the human mitochondrial protein import machinery showing the components of the translocase of the outer (TOM) and inner membrane (TIM23-PAM and TIM22).

B  Human TIMM50 isoforms (top), TIMM50L and TIMM50S, consist of three domains: a mitochondrial targeting sequence (MTS) that directs the protein to mitochondria, a transmembrane domain (TMD), and a presequence binding domain (PBD) that contains an active phosphatase FCP1-like domain. Isoform TIMM50L also contains a nuclear localization signal (NLS). Phylogenetic alignment of the human protein region containing the substitutions found in the affected individual is shown below. Altered residues in the affected subject are boxed in red and its position marked in relation to conserved domains. Abbreviations are as follows: Hs, *H. sapiens*; Pt, *P. troglodytes*; Bt, *B. taurus*; Mm, *M. musculus*; Rn, *R. norvegicus*; Xt, *X. tropicalis*; Dr, *D. rerio*; Dm, *D. melanogaster*; Ce, *C. elegans*; Sc, *S. cerevisiae*.

C  TIMM50S levels in primary and immortalized skin fibroblast from control (C1–4) and mutant *TIMM50* (Pt). SDHA and ETHE1 are shown as loading controls.

D  qPCR quantification of *TIMM50* transcript levels in control and *TIMM50* mutant fibroblasts. Data are shown as mean $\pm$ SD, $n = 4$.

Source data are available online for this figure.

and, unlike the yeast counterpart, a functional serine phosphatase domain (FCP1-like domain), in addition to a predicted N-terminal MTS with low similarity between the two isoforms (Fig 1B) (Guo *et al*, 2004). Import of proteins targeted to the mitochondrial matrix requires an electrochemical mitochondrial membrane potential ($\Delta\Psi$) and ATP hydrolysis, in addition to the action of the presequence translocase-associated motor (PAM) complex. This complex dynamically associates with the TIM23 complex in the so-called TIM23-PAM or TIM23$^{MOTOR}$ complex. Mortalin/HSP9 binds to the precursor proteins and hydrolyzes ATP to pull them inside the organelle, and its attachment to the TIM23 complex is mediated by TIMM44. PAM cochaperones MAGMAS, MCJ, DNAJC19, and GrpE interact with the TIM complex and modulate the ATP-ase activity of mortalin (Sokol *et al*, 2014; Schulz *et al*, 2015). Once in the matrix, mitochondrial processing peptidases (MPP) remove the

presequences, generating the mature form of the imported proteins. A functionally different TIM23 complex is involved in the insertion of MTS-carrying proteins into the IMM, the TIM23$^{SORT}$ complex (Sokol *et al*, 2014). In this case, the PAM complex is replaced by TIMM21, and its interaction with the TIM23 complex is mediated by ROMO1 (Sokol *et al*, 2014; Schulz *et al*, 2015).

Hydrophobic proteins with multiple segments spanning the IMM and no cleavable MTS, such as metabolite carrier proteins, or TIMM23, TIMM22, and TIMM17, require a different translocase, the TIM22 complex (Dudek *et al*, 2013; Sokol *et al*, 2014; Straub *et al*, 2016). A multimeric chaperone complex encompassing TIMM9 and TIMM10, or less frequently TIMM8 and TIMM13, shuttles these proteins from the TOM to the TIM22 complex. Human TIM22 complex consists of the channel-forming subunit TIMM22 and two mammalian specific constituents: the lipid kinase

AGK and TIMM29 (Callegari et al, 2016; Kang et al, 2016; Vukotic et al, 2017).

In humans, only six diseases have been associated with mutations in nuclear genes encoding components of the mitochondrial protein import machinery: three in the TIM23 and two in the TIM22 pathways and one in the disulfide relay system. The deafness–dystonia syndrome (or Mohr–Tranebjaerg syndrome) is an X-linked neurodegenerative disorder caused by mutations of the DDP1 (TIMM8A) gene (Jin et al, 1996; Koehler et al, 1999). This protein is a component of the TIM22 pathway and mediates the insertion of metabolite carriers into the IMM. Mutations in DNAJC19, encoding a mitochondrial PAM chaperone protein, are associated with an autosomal recessive disorder characterized by dilated cardiomyopathy with ataxia (DCMA) (Davey et al, 2006; Ojala et al, 2012). Patients with mutations in GFER (ALR) encoding a mitochondrial intermembrane space protein involved in the regeneration of redox-active disulfide bonds in MIA40 presented autosomal recessive progressive mitochondrial myopathy with congenital cataract, hearing loss, and developmental delay (Di Fonzo et al, 2009). Mutations in MAGMAS, encoding a mitochondrial PAM chaperone protein, are associated with severe spondylodysplastic dysplasia and link mitochondria to the ossification process (Mehawej et al, 2014). Sengers syndrome is an autosomal recessive disorder characterized by congenital cataracts, hypertrophic cardiomyopathy, skeletal myopathy, and exercise intolerance caused by mutations in AGK (Aldahmesh et al, 2012; Mayr et al, 2012; Haghighi et al, 2014), encoding a protein only recently described as a component of the TIM22 complex (Vukotic et al, 2017). Finally, mutations in TIMM50, coding for a core subunit of the TIM23 complex, have been reported in siblings from two unrelated families with severe intellectual disability and seizures, slightly elevated lactate levels, 3-methylglutaconic aciduria and, in one subject, deficiency of mitochondrial complex V (Shahrour et al, 2017).

In this report, we present a thorough molecular dissection of fibroblasts from a patient belonging to a third unrelated family, affected by severe epilepsy and lactic acidosis, with impaired TIM23 complex due to compound heterozygous mutations in TIMM50.

# Results

### Whole exome sequencing

A detailed description of the case report of our patient and a clinical discussion of all the reported cases of TIMM50 mutations are presented in the supplemental material online, including brain magnetic resonance imaging and spectroscopy (Appendix Fig S1). The clinical course, neuroimaging, and biochemical results pointed to an OxPhos-related mitochondrial disorder. Whole exome sequencing and subsequent filtering steps, assuming a recessive trait and focusing on genes encoding mitochondrial proteins, identified two heterozygous variants in TIMM50 (NM_001001563, NP_001001563): a c.335C>A substitution in exon 1, predicted to cause a nonsense mutation p.S112*, and a c.569G>C in exon 3, predicted to produce a missense change (p.G190A; Fig 1B). Sanger sequencing confirmed the two variants in the proband and showed that the two parents were each heterozygous for one of them (Appendix Fig S2A). The nonsense change was not reported in the

ExAc database, while the c.569G>C (rs776019250) was present in only 1 out of > 200,000 alleles. Moreover, the p.G190A aa change occurs in a conserved residue of the protein at the C-terminus of the transmembrane domain and is predicted as "damaging" by different bioinformatics tools.

### TIM-TOM complex protein levels and membrane potential

TIMM50 has been reported in two different isoforms, a nuclear TIMM50L and a mitochondrial TIMM50S, translated from two different transcripts of the same gene. Bioinformatics analyses predicted a mitochondrial targeting signal in both isoforms but also a nuclear localization signal in TIMM50L (Fig 1B). Since only TIMM50S is reported in mitochondria, we will only focus on this isoform and refer to it as TIMM50. The levels of TIMM50 are higher in the control than in patient fibroblasts, both primary and immortalized (Figs 1C and 2A, and Appendix Fig S2B). Indeed, qPCR analysis showed a reduction of about 40% of TIMM50 transcripts in patient fibroblasts, suggesting nonsense-mediated decay of the c.569 G>C (p.S112*) mutant mRNAs (Fig 1D).

TIMM50 is a core component of the mitochondrial protein import machinery, and therefore, we investigated how the mutations present in the patient affected the protein levels of other import components. When the cells were grown in glucose, we detected significantly lower levels of TIMM50 (24–26% of controls) associated with significantly lower levels of the other three components of the TIM23 complex, TIMM17A, TIMM17B, and TIMM23 (3–12%, 14–18% and 15–18%, respectively) and small reduction in DNAJC19 (43–50% of controls; Fig 2A and Appendix Fig S3A). Other components of the PAM complex, channel-forming TIMM22 or subunits of the TOM complex, and its chaperones were not affected, suggesting that the phenotype was exclusively dependent on the TIM23 complex (Fig 2A and Appendix Fig S3A). When fibroblasts were grown in galactose, we observed lower levels of all four TIM23 components and DNAJC19 compared to controls, albeit with higher values than in glucose (Fig 2A and Appendix Fig S3A). This trend was also detected in the other components of the import machinery, with the exception of TOMM40 and TOMM22 that showed lower levels in galactose than in glucose. However, we detected no significant differences between control and patient cells in galactose, except for TOMM34 which was significantly decreased in the patient compared to controls (Fig 2A and Appendix Fig S3A). Furthermore, in the patient cells, the levels of the different proteins are more similar in galactose, where most of them range 0.4–0.7 relative to GAPDH taken as a standard, than in glucose, where TIM23 complex subunits have a value of 0.3 or lower, while all the other proteins are close to 1 (Appendix Fig S3A). By contrast, in control fibroblasts, protein levels were very similar either in glucose or in galactose (Appendix Fig S3A), suggesting that this may represent a characteristic feature of a functionally proficient import machinery. When patient fibroblasts were transduced with TIMM50, comparable levels of TIMM50 to transduced control cells were observed (Fig 2B and Appendix Fig S3B). Furthermore, the levels of complex TIM23 proteins and DNAJC19 were also significantly increased (Fig 2B and Appendix Fig S3B).

Mitochondrial membrane potential is frequently altered in cases of mitochondrial dysfunction, including mutations in TIMM50 that

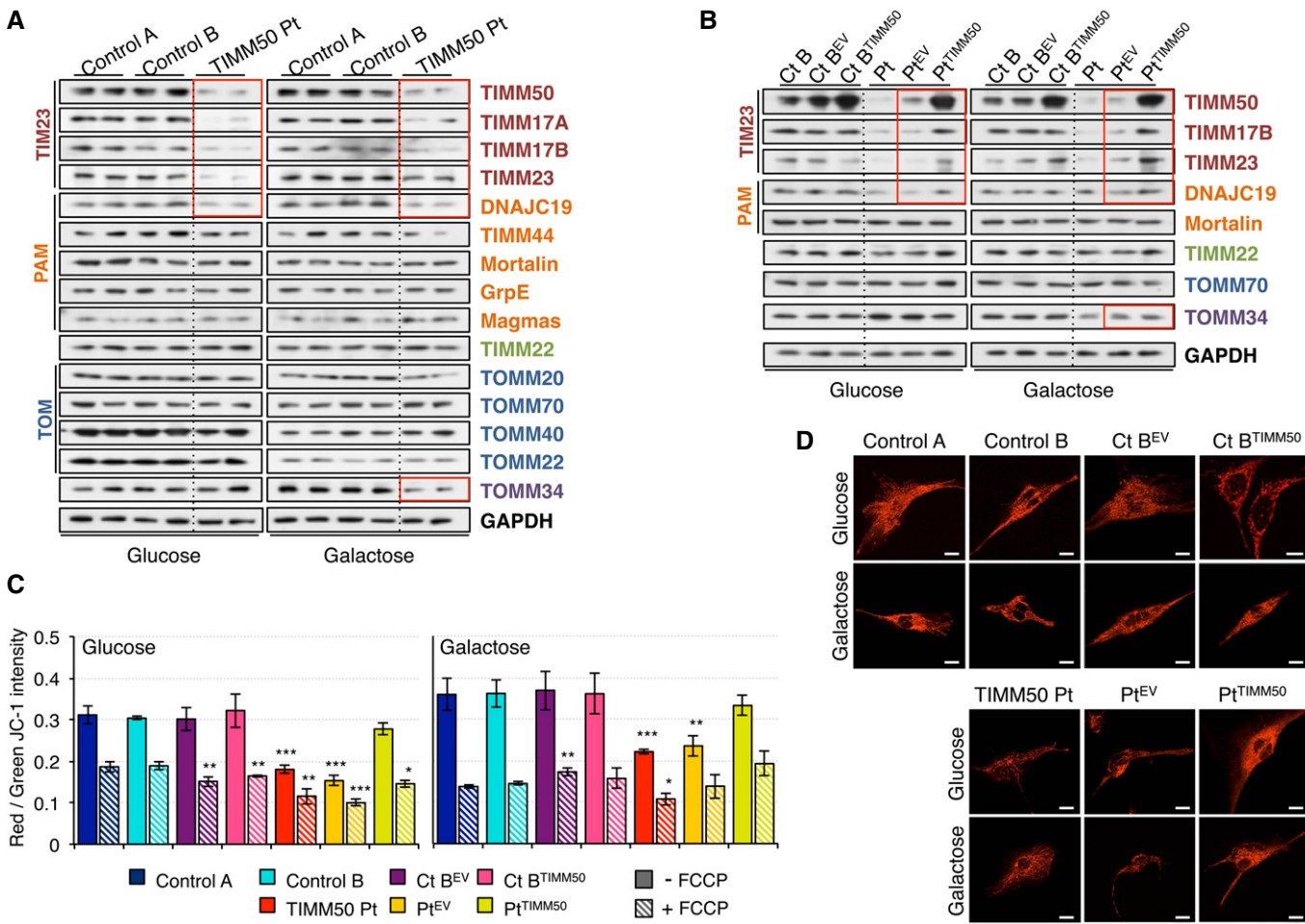

**Figure 2. Decreased TIM23 complex and membrane potential in *TIMM50* mutant fibroblasts.**

A   Western blot of whole cell extracts for different components of the protein import machinery in control and *TIMM50* mutant fibroblasts grown in either glucose- or galactose-containing medium.

B   Western blot of whole cell extracts for different components of the protein import machinery in control and *TIMM50* mutant fibroblasts, transduced with the empty vector (EV) or with wild-type TIMM50 grown in either glucose- or galactose-containing medium.

C   Mitochondrial membrane potential in fibroblasts grown in glucose or galactose using JC-1 staining in untreated or FCCP-treated conditions. Data are shown as mean ± SD, $n = 4$ biological replicates; *$P < 0.05$, **$P < 0.01$, ***$P < 0.001$, Student's unpaired two-tailed $t$-test. Exact $P$-values are reported in Table EV1.

D   TMRM live staining of control and *TIMM50* mutant fibroblasts, transduced with the empty vector (EV) or with wild-type TIMM50 grown in either glucose- or galactose-containing medium. Scale bar corresponds to 10 μm. A digitally enhanced TMRM image from *TIMM50* mutant fibroblasts is shown in Appendix Fig S4B.

Source data are available online for this figure.

make the system leaky, as reported in yeast (Meinecke *et al*, 2006), or alterations in the electron transport chain, impairing proton translocation across the IMM. *TIMM50* mutant fibroblasts displayed significantly lower mitochondrial potential in both glucose and galactose. However, the membrane potential was not completely collapsed since the addition of the uncoupler FCCP resulted in a further decrease in membrane potential in all cases (Fig 2C). Complementation of mutant fibroblasts with transduced TIMM50 increased the membrane potential almost to the level of control cells (Fig 2C). Live staining of fibroblasts with the membrane potential-dependent dye TMRM confirmed the lower membrane potential in the patient fibroblasts grown in glucose, with mild improvement when grown in galactose and a significant improvement when transduced with TIMM50 (Fig 2D and Appendix Fig S4A and B).

**Import through the TIM23 and the TIM22 complexes**

In order to assess whether the *TIMM50* mutations affect import pathways, we synthesized TFAM and AAC1 polypeptides in a cell-free system and performed protein import experiments into mitochondria isolated from control and patient fibroblasts (Fig 3). TFAM import into mitochondria occurs in a membrane potential-dependent way (Fig 3A, Reyes *et al*, 2011), and as a matrix protein with a cleavable MTS, the import of TFAM is carried out by the TIM23 pathway. Mitochondria from control fibroblasts were able to import TFAM over time while its import in patient mitochondria was severely impaired (Fig 3A and B). AAC1 import into mitochondria also occurs in a membrane potential-dependent way but using the TIM22 pathway instead (Fig 3A, Vukotic *et al*, 2017).

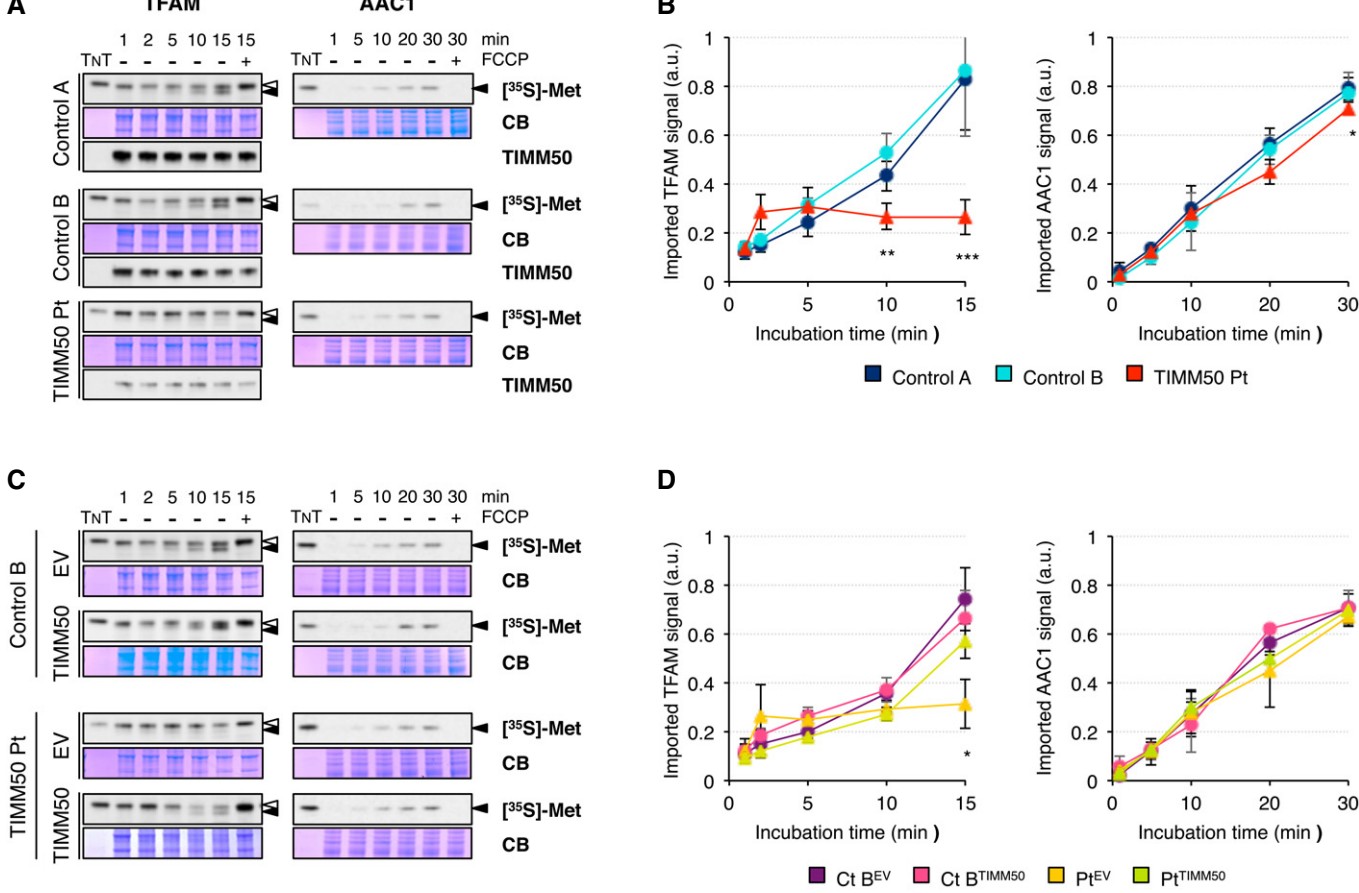

**Figure 3.  TIM23 but not TIM22-dependent import is impaired in *TIMM50* mutant fibroblasts.**

A    *In organello* import of the precursors of TFAM and AAC1 into isolated mitochondria from control and *TIMM50* mutant fibroblasts in the absence or presence of the uncoupler FCCP. The radiolabeled precursors were incubated with isolated mitochondria for the indicated times followed by trypsin treatment only in the case of AAC1. Coomassie Blue (CB) staining was used as loading control and the radiolabeled protein (TNT) as reference for quantification. The position of the precursor (empty arrowheads) and the mature protein (filled arrowhead) are also shown. Western blot for TIMM50 is shown as cell line control.

B    Quantification of TFAM and AAC1 import from control and *TIMM50* mutant fibroblasts expressed as fraction of the input radiolabeled precursor converted into mature protein and normalized to Coomassie Blue signal. Data are shown as mean ± SD, n = 3 biological replicates; *P < 0.05, **P < 0.01, ***P < 0.001 grouping both controls together, Student's unpaired two-tailed *t*-test. Exact *P*-values are reported in Table EV1.

C    *In organello* import of the precursors of TFAM and AAC1 into isolated mitochondria from control and *TIMM50* mutant fibroblasts transduced with the empty vector (EV) or with wild-type TIMM50 in the absence or presence of the uncoupler FCCP as described in (A). The position of the precursor (empty arrowheads) and the mature protein (filled arrowhead) are also shown.

D    Quantification of TFAM and AAC1 import from control and *TIMM50* mutant fibroblasts transduced with the empty vector (EV) or with wild-type TIMM50 expressed as fraction of the input radiolabeled precursor converted into mature protein and normalized to Coomassie Blue signal. Data are shown as mean ± SD, n = 3 biological replicates; *P < 0.05. Exact *P*-values are reported in Table EV1.

Source data are available online for this figure.

Notably, the decrease in TIM23 complex proteins had only a minor effect (10–12% decrease at 30 min) in the import of this protein in patient mitochondria, which contain normal levels of TIMM22 (Fig 3A and B), providing evidence that the residual mitochondrial membrane potential observed in patient fibroblasts is enough to allow import of proteins through the IMM. In agreement with the import results, we observed lower steady-state levels of TFAM in patient fibroblasts and no significant difference for AAC1 (Appendix Fig S4C). Mitochondria from patient fibroblasts transduced with TIMM50 were able to import TFAM in a similar fashion to transduced control (Fig 3C and D), demonstrating not only that TIMM50 levels were increased but also that the function of

the TIM23 complex was restored. No significant changes were observed for AAC1 import.

## Changes in OxPhos proteins and oxygen consumption

Next, we investigated how the import impairment detected in patient mitochondria would affect structural components of the electron transport chain. OxPhos constituents of all five complexes were analyzed by SDS–PAGE and Western blot. Patient fibroblasts grown in glucose presented significantly lower steady-state levels of all surveyed subunits of complex II and complex IV and one complex I subunit, while no difference was observed for either

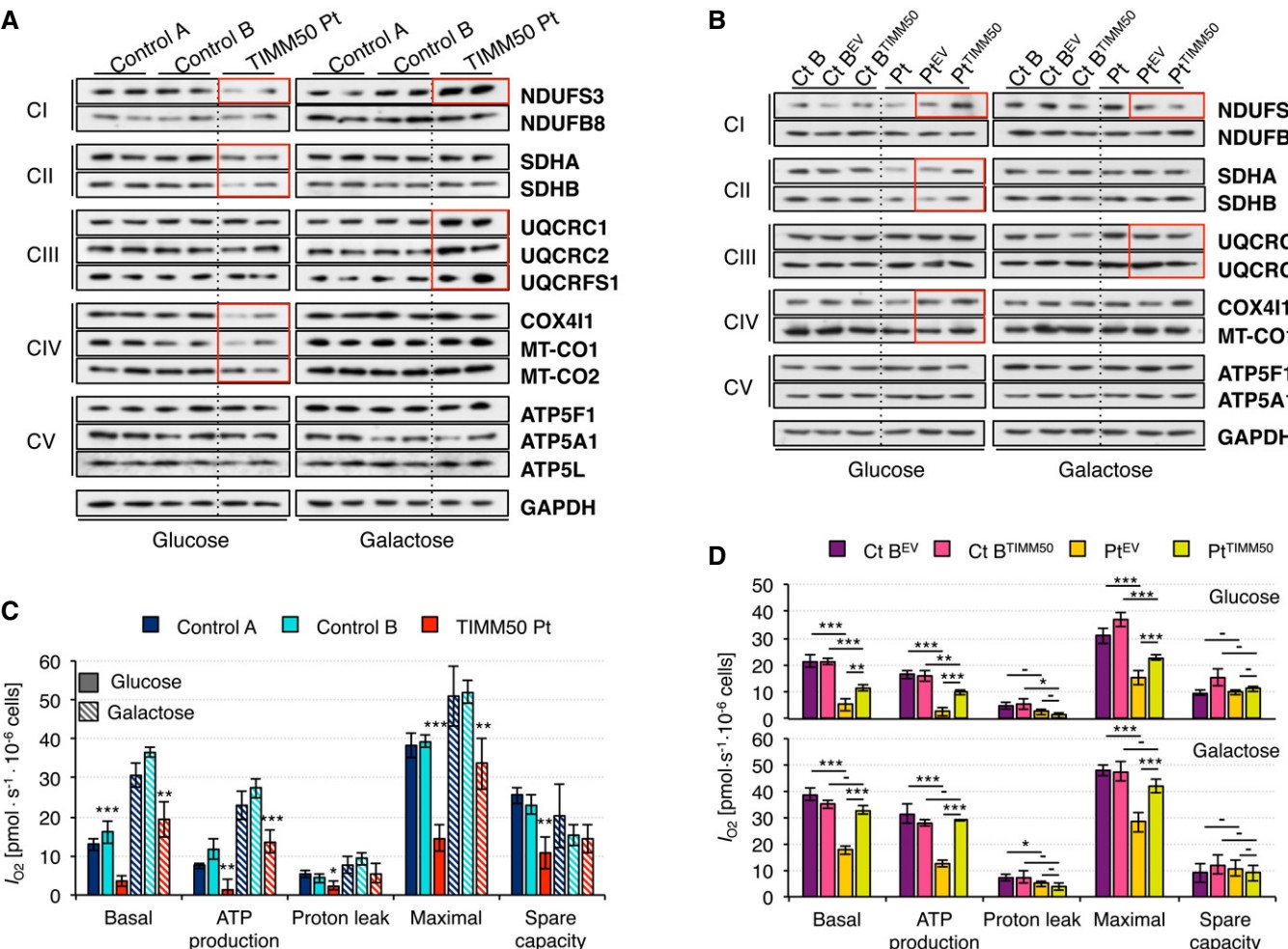

**Figure 4. Altered OxPhos protein level and respiration in *TIMM50* mutant fibroblasts.**

A   Western blot for representatives of all five respiratory complexes in control and *TIMM50* mutant fibroblasts grown in glucose or galactose. Two biological replicates are shown for each cell line and condition. Significant changes in protein levels in *TIMM50* mutant fibroblasts compared to controls have been boxed in red. The quantification is shown in Appendix Fig S3A.

B   Western blot for representatives of all five respiratory complexes in control and *TIMM50* mutant fibroblasts, transduced with the empty vector (EV) or with wild-type TIMM50 grown in glucose or galactose. Two biological replicates were obtained for each cell line and condition. Significant changes in protein levels in *TIMM50* mutant fibroblasts complemented with TIMM50 compared to EV have been boxed in red. The quantification is shown in Appendix Fig S3B.

C   Oxygen consumption measurements in control and *TIMM50* mutant fibroblasts grown in glucose or galactose. Values of basal and maximal respiration along with ATP production-dependent, proton leak respiration, and spare capacity are shown. Data are shown as mean ± SD, $n = 4$ biological replicates; *$P < 0.05$, **$P < 0.01$, ***$P < 0.001$ grouping both controls together, Student's unpaired two-tailed $t$-test. Exact $P$-values are reported in Table EV1.

D   Oxygen consumption measurements in control and *TIMM50* mutant fibroblasts, transduced with the empty vector (EV) or with wild-type TIMM50 grown in glucose or galactose. Values of basal and maximal respiration along with ATP production-dependent, proton leak respiration, and spare capacity are shown. Data are shown as mean ± SD, $n = 4$ biological replicates; *$P < 0.05$, **$P < 0.01$, ***$P < 0.001$, Student's unpaired two-tailed $t$-test. Exact $P$-values are reported in Table EV1.

Source data are available online for this figure.

complex III or complex V subunits (Fig 4A and Appendix Fig S5A). However, the decrease in MT-CO1 and MT-CO2 must be secondary to the decrease of nuclear-encoded complex IV subunits and/or to a reduction in the mitochondrial transcription/translation machinery, since both proteins are encoded by the mitochondrial DNA and therefore, they do not require to be imported. Growth in galactose showed an overall increase of OxPhos proteins, in agreement with increased biogenesis. Furthermore, no differences between patient and control fibroblasts were observed in galactose for either complex II or complex IV. In addition, all

complex III and one complex I subunit showed even higher values in patient than in control fibroblasts in galactose (Fig 4A and Appendix Fig S5A). These results are concordant with an apparent improvement of protein levels of the import machineries in patient fibroblasts grown in galactose (Fig 2A and Appendix Fig S3B). Complementation of patient fibroblasts by TIMM50 transduction increased the steady-state levels of subunits of complexes I, II, and IV in glucose and decreased the levels of complexes I and III in galactose to comparable values to transduced controls (Fig 4B and Appendix Fig S5B).

Since we observed significant changes in protein levels for some complex subunits, we then investigated what effect this may have on mitochondrial respiration, by measuring oxygen consumption, $I_{O2}$. When grown in glucose, patient fibroblasts showed significantly lower values than control in every single category: basal $I_{O2}$, ATP-dependent, proton leak, maximal, and spare $I_{O2}$ capacity (Fig 4C). Notably, ATP-dependent respiration was actually extremely low in the patient fibroblasts, despite the normal levels of all complex V subunits we analyzed (Fig 4A and Appendix Fig S5A). Galactose-grown fibroblasts displayed higher values in basal $I_{O2}$, ATP-dependent, and maximal respiration than those grown in glucose. This effect is likely due to a metabolic shift from glycolysis to OxPhos, and it is also in agreement with the observed increase in protein levels of respiratory chain complex subunits (Fig 4A and Appendix Fig S5A). As a consequence, spare capacity is reduced compared to glucose in all cases. Nevertheless, respiration was still lower in patient than in control fibroblasts. However, ATP production in patient fibroblasts reached values similar to those found in control cells grown in glucose, which again points to an apparent bioenergetics improvement of patient cells grown in galactose. Basal, maximal, and ATP-dependent oxygen consumption were significantly increased in patient fibroblasts transduced with TIMM50 compared to those transduced with empty vector when cells were grown in both glucose and galactose. However, oxygen consumption values were still lower compared to those of transduced controls in glucose, whereas they were similar in galactose (Fig 4D). These results are also in agreement with the changes detected in protein levels of respiratory chain complex subunits in transduced patient fibroblasts (Fig 4B and Appendix Fig S5B).

## ROS production

A possible direct consequence of dysfunctional OxPhos is the increase of reactive oxygen species (ROS). ROS production, as measured by intracellular oxidation of DCFDA, was higher in cells grown in glucose than galactose (Fig 5A). When cells were left untreated, patient fibroblasts only showed a marginal increase of ROS levels at later time points. However, the pre-incubation with exogenous hydrogen peroxide clearly enhanced the differences and revealed significantly higher levels of ROS in the patient fibroblasts in both glucose and galactose. To further verify these results, we investigated the steady-state levels of two proteins related to oxidative stress response and in particular to superoxide anion $O_2^-$ detoxification, SOD2 and ACO2. While both proteins were significantly upregulated in the patient fibroblasts and the effect was more pronounced in galactose, SOD2 presented the highest increase compared to control cells (Fig 5B and C). This increase was reversed when patient fibroblasts were transduced with TIMM50 (Appendix Fig S6A–C).

Decreased mitochondrial potential and increased ROS production have been associated with increased selective mitochondrial degradation or mitophagy. Thus, we analyzed the levels of VDAC1, a protein whose ubiquitination by Parkin can trigger ROS-mediated mitophagy, as well as the ubiquitin-binding adaptor protein p62 and the phagophore protein LC3. Patient fibroblasts grown in glucose showed a significantly higher level of VDAC1 and lipidated LC3-II in concomitance with a significant decrease of p62, resulting in higher LC3-II/p62 ratio (Fig 5D and E) and pointing toward increased

mitophagy. Such differences were not observed when fibroblasts were grown in galactose, being the increase in LC3-II in all samples the most prominent change compared to the results obtained in glucose (Fig 5D and E). Transduction of patient fibroblasts with TIMM50 restored levels of VDAC1 and p62 and to some extent lipidated LC3-II, to comparable levels to transduced control, in agreement with the observed decrease in levels of SOD2 and ACO2 (Appendix Fig S6A and D–E).

## Cell viability

In order to assess what impact mutations in *TIMM50* and the subsequent mitochondrial alterations may have in the cell as a whole, we measured cell size and cell growth in glucose- and galactose-containing media (Fig 6). Patient fibroblasts were significantly smaller than control in both glucose and galactose but their size was increased in both conditions when they were transduced with TIMM50 (Fig 6A). In addition, their growth rate in glucose was 50% slower than controls but still able to reach confluency albeit with a decrease in viability (Fig 6B and Appendix Fig S7). Transduction with TIMM50 significantly improved patient fibroblast growth and viability in glucose (Fig 6B and Appendix Fig S7). However, when the cells were grown in galactose, a very limited growth was observed (Fig 6B) despite all the apparent biochemical improvements shown in the previous experiments. This growth defect was reversed by transduction with TIMM50. This result could be due to either lower division rate or increased cell death in galactose. Therefore, we measured apoptosis in glucose and galactose conditions. We observed no significant increase in apoptosis in glucose; however, there was a fivefold increase in apoptotic cells when patient fibroblasts were grown in galactose medium while no such change was detected in controls; importantly, this phenotype was lost after transduction with TIMM50 (Fig 6C and D).

### *RNASEH1* and *ISCU* patient fibroblasts

Immortalized fibroblasts derived from patients with mutations in *RNASEH1*, coding for a protein involved in mtDNA maintenance (Reyes *et al*, 2015) or *ISCU*, encoding a scaffold protein for the assembly of iron-sulfur (Fe-S) clusters (Legati *et al*, 2017) have been studied as additional cell models of mitochondrial disease in order to establish if the reported changes were unique to mutant *TIMM50* fibroblasts or shared with other disease cell lines. Steady-state levels of the components of the mitochondrial protein import machinery in *RNASEH1*, and *ISCU* mutant fibroblasts were not very different from controls (Fig 7A and Appendix Fig S8A and B) except for TOMM34 when cells were grown in galactose, as also observed in *TIMM50* mutant fibroblasts (Appendix Fig S8A). However, TFAM and AAC1 mitochondrial import was not significantly affected in these two cell lines (Appendix Fig S8C) despite both of them displayed significantly lower mitochondrial membrane potential than control (Fig 7B).

Analysis of OxPhos constituents of all five complexes revealed that significant changes compared to control were only detected for SDHB and COX4I1 in *ISCU* mutant fibroblasts grown in glucose and MT-CO1 in both *RNASEH1* and *ISCU* mutant fibroblasts grown in galactose (Fig 7C and Appendix Fig S9). The most dramatic changes detected in *TIMM50* mutant fibroblasts, e.g., lower levels of

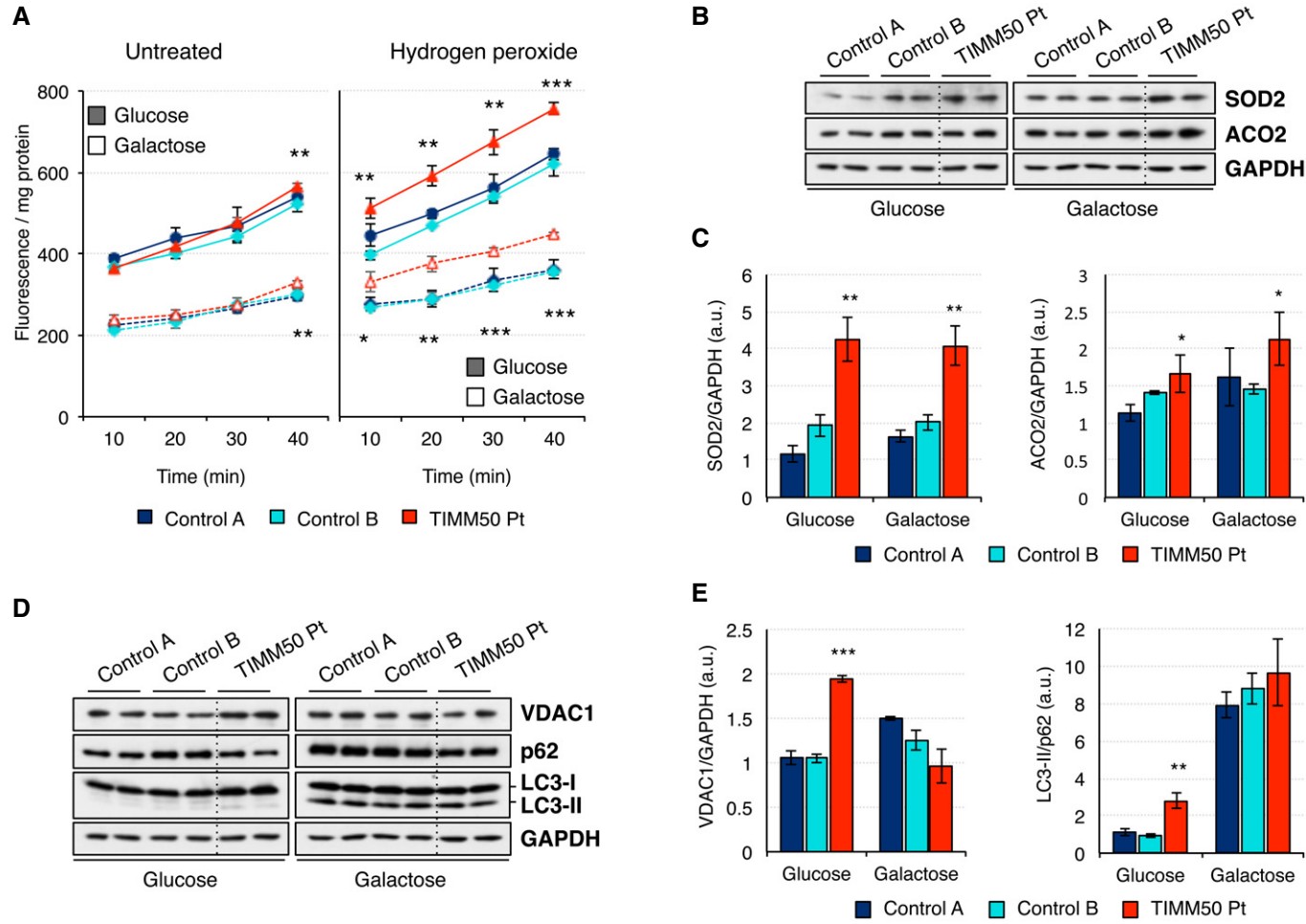

**Figure 5. Increased ROS production and mitophagy in *TIMM50* mutant fibroblasts.**

A   ROS production as measured by intracellular oxidation of the cell-permeant probe DCFDA in control and *TIMM50* mutant fibroblasts grown in glucose or galactose. Measurements were made in untreated cells and after treatment with 250 μM hydrogen peroxide for 30 min. Data are shown as mean ± SD, n = 3 biological replicates; *$P < 0.05$, **$P < 0.01$, ***$P < 0.001$ grouping both controls together, Student's unpaired two-tailed *t*-test. Exact *P*-values are reported in Table EV1.

B   Steady-state levels of ROS-related proteins SOD2 and ACO2 in control and *TIMM50* mutant fibroblasts growing in either glucose or galactose were assessed by Western blot using GAPDH as internal control.

C   Quantification of the steady-state levels of SOD2 and ACO2 in control and *TIMM50* mutant fibroblasts growing in either glucose or galactose. Data are shown as mean ± SD, n = 4 biological replicates; *$P < 0.05$, **$P < 0.01$ grouping both controls together, Student's unpaired two-tailed *t*-test. Exact *P*-values are reported in Table EV1.

D   Steady-state levels of mitophagy-related proteins VDAC1, p62, and LC3-II in control and TIMM50 mutant fibroblasts growing in either glucose or galactose were assessed by Western blot using GAPDH as internal control. GAPDH is from the same blot as panel (B).

E   Quantification of the steady-state levels of VDAC1, p62, and LC3-II in control and *TIMM50* mutant fibroblasts growing in either glucose or galactose. Data are shown as mean ± SD, n = 4 biological replicates; **$P < 0.01$, ***$P < 0.001$ grouping both controls together, Student's unpaired two-tailed *t*-test. Exact *P*-values are reported in Table EV1.

Source data are available online for this figure.

NDUFS3 in glucose or the increase of NDUFS3 and complex III components in galactose, were not present in the other two cell lines (Fig 7C and Appendix Fig S9). In agreement with these results, oxygen consumption, $I_{O_2}$, in *RNASEH1* mutant fibroblasts was similar to controls in most cases and only slightly lower in *ISCU* mutant fibroblasts grown in glucose, but always higher than in *TIMM50* mutant fibroblasts. However, both cell lines showed moderately lower oxygen consumption than controls when grown in galactose, albeit significantly higher than *TIMM50* mutant fibroblasts (Fig 7D). As a consequence, the significant increase in proteins related to

oxidative stress response, SOD2 and ACO2, that we observed in *TIMM50* mutant fibroblasts was not present in *RNASEH1* and *ISCU* mutant fibroblasts (Fig 7E and Appendix Fig S10), and indeed, the levels of ACO2 in *ISCU* mutant fibroblasts were actually lower then controls as ACO2 is an iron-sulfur protein dependent on the activity of the ISCU protein. By contrast, the levels of a protein involved in ROS-mediated mitophagy, VDAC1, were significantly higher in *ISCU* mutant fibroblasts than in controls but not so high as in *TIMM50* mutant fibroblasts (Fig 7E and Appendix Fig S10). The only common feature in all three patient fibroblast cell lines was the

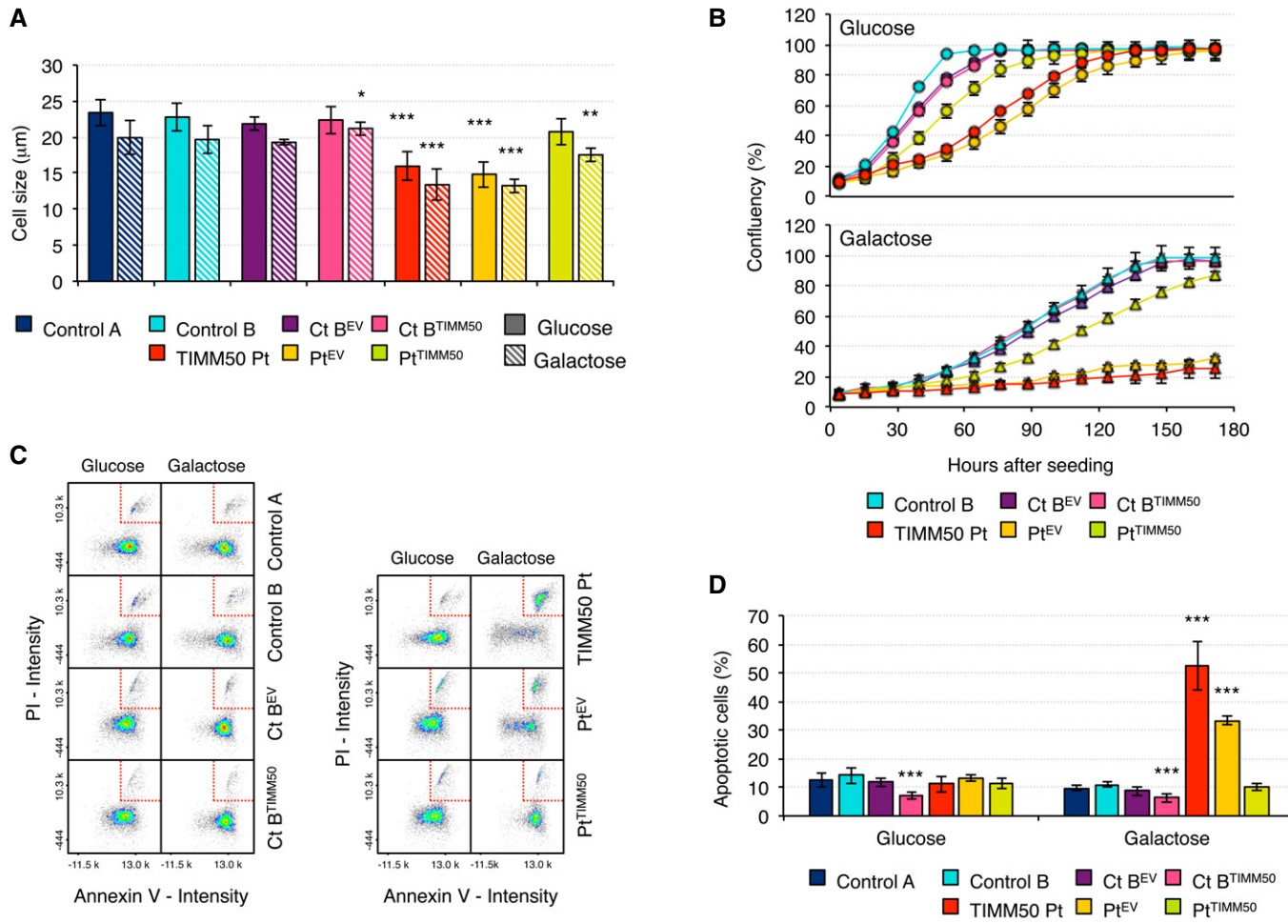

**Figure 6. Cell size, cell growth, and apoptosis in *TIMM50* mutant fibroblasts.**

A  Cell size in control and *TIMM50* mutant fibroblasts, transduced with the empty vector (EV) or with wild-type TIMM50 grown in glucose or galactose. Data are shown as mean ± SD, n = 5 biological replicates; *P < 0.05, **P < 0.01, ***P < 0.001, Student's unpaired two-tailed t-test. Exact P-values are reported in Table EV1.

B  Growth curves of control and *TIMM50* mutant fibroblasts, transduced with the empty vector (EV) or with wild-type TIMM50 grown in glucose or galactose. Cell growth was monitored continuously by the Incucyte live cell imager (Essen Bioscience). One of the three independent experiments carried out is presented. Data are shown as mean of three technical replicates ± SD.

C  Plot of Annexin V intensity against propidium iodide signal in control and *TIMM50* mutant fibroblasts, transduced with the empty vector (EV) or with wild-type TIMM50 grown in glucose or galactose. Cells in the late stage of apoptosis are boxed in the top right corner.

D  Quantification of the number of late apoptotic cells in control and *TIMM50* mutant fibroblasts, transduced with the empty vector (EV) or with wild-type TIMM50 grown in glucose or galactose. Data are shown as mean ± SD, n = 5 or 4 biological replicates in the case of transduced cells; ***P < 0.001, Student's unpaired two-tailed t-test. Exact P-values are reported in Table EV1.

significant low levels of p62 in glucose growth, that is persistent in galactose growth only in the case of *TIMM50* mutant fibroblasts (Fig 7E and Appendix Fig S10).

*ISCU* but not *RNASEH1* mutant fibroblasts were smaller than controls, similar to *TIMM50* mutant fibroblasts both in glucose- and galactose-containing media (Appendix Fig S11A). Likewise, *RNASEH1* mutant fibroblasts displayed a marginal growth defect compared to controls, while a growth defect was observed for *ISCU* and *TIMM50* mutant fibroblasts, with a severe impairment in galactose, albeit consistently less pronounced in *ISCU* vs. *TIMM50* mutant fibroblasts (Appendix Fig S11B). Cell viability was not affected in *RNASEH1* and *ISCU* mutant fibroblasts as observed in *TIMM50* mutant fibroblasts (Appendix Fig S11C) nor was the number of

apoptotic cells when cultured in galactose media (Fig 7F), a feature that remains specific of *TIMM50* mutant fibroblasts (Fig 6D).

## Discussion

Two isoforms of TIMM50 have been described: a 353 aa-long mitochondrial isoform TIMM50S and a 456 aa-long nuclear isoform TIMM50L (Xu *et al*, 2005). The official nomenclature of mutations refers to the longer species, which has an alternative translational start sequence adding 103 aa to the N-terminus. The nonsense mutation p.S112* introduces a stop codon after 112 or 9 aa from the initial methionine of TIMM50L and TIMM50S, respectively, and

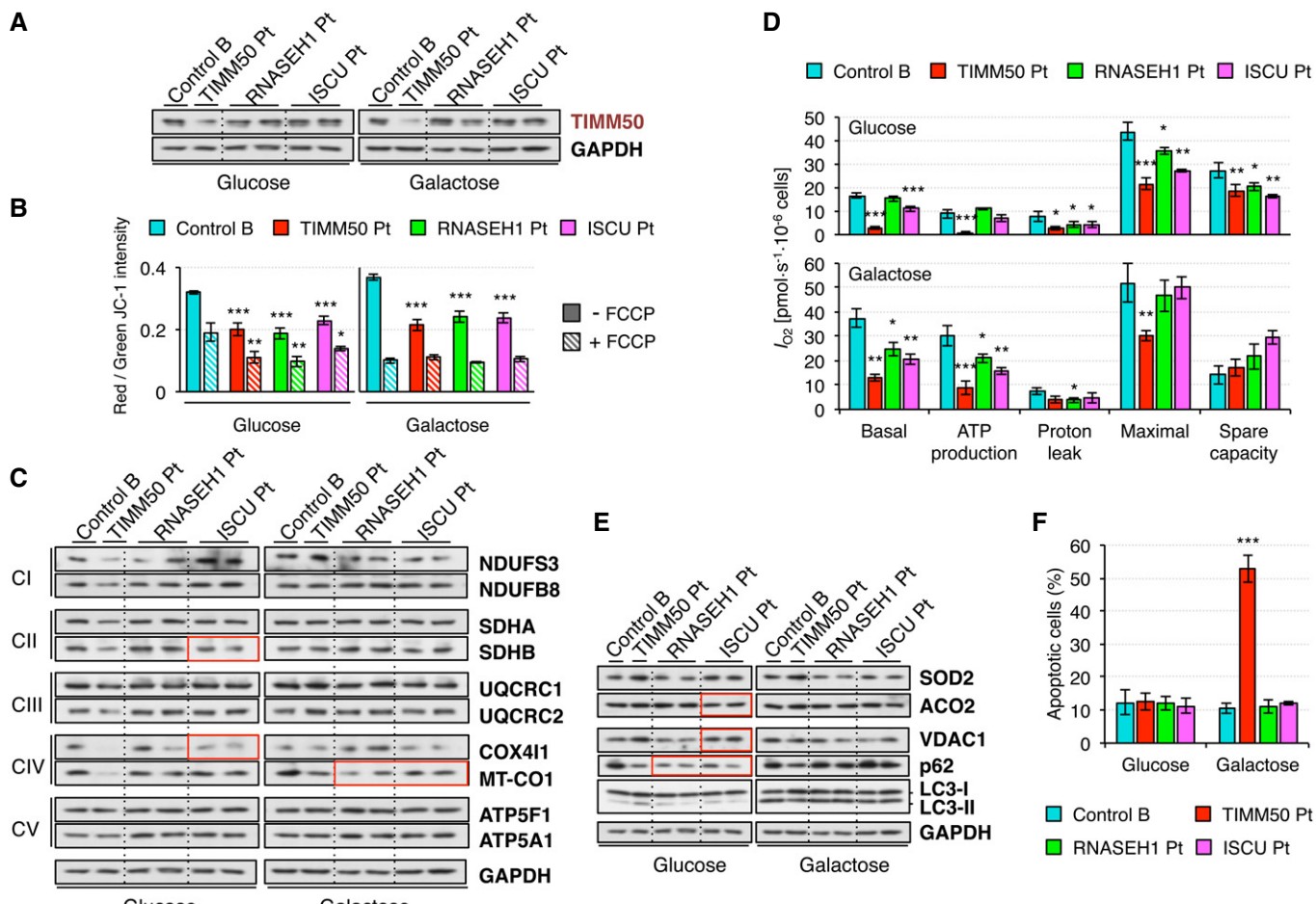

**Figure 7. Characterization of *RNASEH1* and *ISCU* mutant fibroblasts.**

A   Western blot of whole cell extracts for TIMM50 and GAPDH in control and *TIMM50*, *RNASEH1*, and *ISCU* mutant fibroblasts grown in either glucose- or galactose-containing medium. Other components of the protein import machinery are shown in Appendix Fig S8A and their quantification in Appendix Fig S8B.

B   Mitochondrial membrane potential in control and *TIMM50*, *RNASEH1*, and *ISCU* mutant fibroblasts grown in glucose or galactose using JC-1 staining in untreated or FCCP-treated conditions. Data are shown as mean ± SD, $n$ = 4 biological replicates; *$P$ < 0.05, **$P$ < 0.01, ***$P$ < 0.001, Student's unpaired two-tailed $t$-test. Exact $P$-values are reported in Table EV1.

C   Western blot for representatives of all five respiratory complexes in control and *TIMM50*, *RNASEH1*, and *ISCU* mutant fibroblasts grown in glucose or galactose. GAPDH is from the same blot as panel (A). Quantification based on two biological replicates for *RNASEH1* and *ISCU* mutant fibroblasts is shown in Appendix Fig S9. Significant changes in protein levels in *RNASEH1* and *ISCU* mutant fibroblasts compared to controls have been boxed in red.

D   Oxygen consumption measurements in control and *TIMM50*, *RNASEH1*, and *ISCU* mutant fibroblasts grown in glucose or galactose. Values of basal and maximal respiration along with ATP production-dependent, proton leak respiration, and spare capacity are shown. Data are shown as mean ± SD, $n$ = 4 biological replicates; *$P$ < 0.05, **$P$ < 0.01, ***$P$ < 0.001, Student's unpaired two-tailed $t$-test. Exact $P$-values are reported in Table EV1.

E   Steady-state levels of ROS-related proteins SOD2 and ACO2 and mitophagy-related proteins VDAC1, p62, and LC3 in control and *TIMM50*, *RNASEH1*, and *ISCU* mutant fibroblasts growing in either glucose or galactose were assessed by Western blot using GAPDH as internal control. GAPDH is from the same blot as panel (A). Quantification is shown in Appendix Fig S10. Significant changes in protein levels in *RNASEH1* and *ISCU* mutant fibroblasts compared to controls have been boxed in red.

F   Quantification of the number of late apoptotic cells in control and *TIMM50*, *RNASEH1*, and *ISCU* mutant fibroblasts grown in glucose or galactose. Data are shown as mean ± SD, $n$ = 4 or more biological replicates; ***$P$ < 0.001, Student's unpaired two-tailed $t$-test. Exact $P$-values are reported in Table EV1.

Source data are available online for this figure.

could be responsible for the lower levels of protein detected in patient fibroblasts compared to control, due to nonsense-mediated decay. The strong reduction in TIMM50 amount also suggests partial instability of the mutant protein with the p.G190A missense mutation. We showed that lower levels of TIMM50S lead to instability of the TIM23 complex and as a consequence a decrease in the other components of the complex, TIMM23 and TIMM17A/B, and

DNAJC19 with drastic consequences for the TIM23 import pathway. On the other hand, the missense change p.G190A (p.G87A in the mitochondrial TIMM50S sequence) hits the second last amino acid of the transmembrane domain. Glycine is a very flexible residue, and it has been replaced by alanine, which normally creates a kink in the structure. This change may alter the capacity of the presequence binding domain (PBD) of TIMM50 to interact with the other

components of the TIM23 complex, and as a consequence, the channel may not close tightly at resting state, leading to the dissipation of membrane potential that we observed in the patient fibroblasts, which was rescued by transduction with TIMM50. Indeed, yeast Tim50, and more precisely the PBD, has been shown to induce the closure of the Tim23 channel and hence maintain mitochondrial membrane potential (Meinecke *et al*, 2006). Moreover, Tim50 PBD is responsible for the precursor handover from Tom22 to Tim23, causing conformational changes that open the Tim23 channel and allow import progression (Qian *et al*, 2011; Schulz *et al*, 2011, 2015). Thus, mutations affecting the flexibility of the PDB and/or its ability to interact either with the presequence or other import subunits are expected to directly affect import rate in the TIM23 pathway. Indeed, we have shown that import of TFAM is severely impaired in patient fibroblasts but restored in TIMM50 transduced cells, whereas import of AAC1 is as efficient as in controls. TFAM is imported via the TIM23 complex, while ACC1 requires active TIM22 complex and the import of both proteins is dependent on membrane potential (Reyes *et al*, 2011; Vukotic *et al*, 2017). Hence, residual membrane potential observed in the patient may be enough to allow the import of both proteins, at least in isolated mitochondria. Indeed, *RNASEH1* and *ISCU* mutant fibroblasts display similar low membrane potential to *TIMM50* mutant fibroblasts but no effect on mitochondrial import was observed. While normal levels of TIM22 complex result in efficient ACC1 import, the decrease in TIM23 complex and its lower presequence handover efficiency can be responsible for the impaired TFAM import. In *Trypanosoma brucei*, Tim50 knock-down also resulted in less efficient import of TIM23-imported MRP2 and COIV, with no significant effect on ACC1 import, despite a 25% decrease in mitochondrial membrane potential (Duncan *et al*, 2013). Likewise, partial depletion of Tim50 in yeast resulted in mtHsp60 import impairment without any effect on ACC (Yamamoto *et al*, 2002). Although a pathogenic mutation in human *TIMM50* failed to be associated with import impairment (Shahrour *et al*, 2017), we noticed that Shahrour *et al* (2017) used yeast as an *in vivo* model system to validate the mutation, despite the fact that human TIMM50 cannot rescue the phenotype of yeast ΔTim50 and the global conservation between these two proteins is very poor.

Impairment of the mitochondrial import system is expected to have dramatic consequences for the cell, and indeed, we have observed significant reduction of some OxPhos subunits in glucose-grown cells that can be rescued by transduction with TIMM50 and is only partially present in *ISCU* but not in *RNASEH1* mutant fibroblasts, most likely due to alterations in iron-sulfur metabolism in the former. However, not all complexes are equally affected, and significant differences were only detected for complexes I, II, and IV. The use of MitoBloCK-6, a small molecule that impairs mitochondrial protein import, has also been reported to produce the same result in HeLa cells (Quiros *et al*, 2017). Although this differential effect on the level of OxPhos subunits is poorly understood, it could be explained as a consequence of different factors, as follows. First, (i) protein steady-state levels are a balance between new synthesis and degradation, and therefore, the half-life of a protein can be altered in a protein-specific way when synthesis, or import, as in this case, is impaired. Second, (ii) submitochondrial localization requires the use of either TIM23-PAM for matrix proteins or TIM23^SORT for inner membrane proteins, and in the latter pathway, interaction with TIMM21 leads to their stabilization in subcomplexes, thus

preventing their degradation, as it has been shown in HEK293 cells for NDUFB8, which interacts with TIMM21, compared to NDUFS6, which does not (Mick *et al*, 2012). These results are in agreement with those of Tim50 depletion in yeast, where import of matrix proteins was strongly affected, whereas inner membrane-targeted proteins were much less dependent on integrity of Tim50 (Schendzielorz *et al*, 2017). Third, (iii) protein import may be efficient even in conditions of low TIM23 complex and/or absence of mitochondrial membrane potential, as already described for small single-spanning inner membrane complex V (ATP5I/Su e) and for a complex III (Qcr10) subunits in yeast (Turakhiya *et al*, 2016). In yeast depleted of Tim50, import of some mitochondrial matrix proteins is impaired in the presence of only minor changes in the mitochondrial membrane potential, while other proteins are hardly affected except when the membrane potential is severely decreased, being this feature not dependent on the presequence but rather on the mature portion of the protein (Schendzielorz *et al*, 2017). Finally, (iv) some proteins may be able to be imported using alternative pathways to overcome import defects, as suggested by the yeast phosphate carrier, which can be imported by either the TIM22 or the TIM23 pathways (Yamano *et al*, 2005). As already mentioned, the import deficiency detected in our *TIMM50* mutant cells results in lower levels of some OxPhos proteins causing a reduction of both basal and maximal mitochondrial respiration, as well as ATP-dependent respiration, and transduction with TIMM50 substantially rescued this deficiency, albeit not reaching control levels. HeLa cells treated with MitoBloCK-6 also displayed a decrease in mitochondrial respiration (Quiros *et al*, 2017), and Ttm50 depletion in *Drosophila* leads to reduced respiratory activity (Sugiyama *et al*, 2007). By contrast, *RNASEH1* and *ISCU* mutant fibroblasts showed no or a small decrease in mitochondrial respiration. OxPhos deficiency has also been associated with ROS production and, accordingly, we detected a slight increase in ROS in untreated cells that becomes more apparent after exposure to hydrogen peroxide. MitoBloCK-6-treated HeLa cells also display elevated ROS production (Quiros *et al*, 2017). In agreement with these results, we found increased levels of two proteins, SOD2 and ACO2, involved in the conversion of superoxide anion into hydrogen peroxide, which will be further eliminated by catalase and glutathione system. VDAC1 also plays a role in ROS detoxification by mediating the superoxide anion release from mitochondria to cytosol, where it is a substrate of cytosolic SOD1 (Han *et al*, 2003). This may explain why VDAC1 is increased in patient cells. Significant upregulation of VDAC1 has also been reported in *T. brucei* Tim50 knock-down, whereas Tim50 overexpression leads to its downregulation (Duncan *et al*, 2013). Moreover, VDAC1 can also be ubiquitinated by Parkin, triggering ROS-mediated mitophagy (Geisler *et al*, 2010). The reduction of p62 and increase of the lipidated form of LC3 are markers of increased autophagic flux, and both features are present in the patient cells, pointing to a slight increase in mitophagy. All these metabolic changes can be attributed to the lack of TIMM50 since patient fibroblasts complemented with TIMM50 presented values comparable to controls in all cases and they were not observed in *RNASEH1* and *ISCU* mutant fibroblasts, with the exception of lower p62 levels. As a consequence of these metabolic alterations, *TIMM50* mutant fibroblasts displayed a clear cellular phenotype characterized by smaller cell size and slower growth rate that was, at least, partially reverted by

transduction with TIMM50. Several lines of evidence support these observations: TIMM50 stable silencing in human breast cancer cell lines suppresses cell proliferation (Gao *et al*, 2016); Ttm50-depleted *Drosophila* has a diminutive body due to a reduction in both size and number of cells caused by cell growth and proliferation defects (Sugiyama *et al*, 2007); and Tim50 knock-down in *T. brucei* produces a slow growth phenotype (Duncan *et al*, 2013). Haploinsufficiency of Tim23 in the heterozygous Tim23 knockout mouse has also been correlated with a markedly reduced life span (Ahting *et al*, 2009). However, this cellular phenotype is not unique to TIMM50 mutations, as we can also observe it in *ISCU* mutant fibroblasts, albeit not so pronounced.

It has long been known that cells growing in glucose utilize both glycolysis and OxPhos for ATP production, with fibroblasts mainly relying on a glycolytic metabolism, whereas cells growing in galactose are almost entirely dependent on OxPhos (Reitzer *et al*, 1979). The underlying reason is that glycolytic metabolism of glucose yields two net ATP molecules, while no net ATP is produced from galactose, forcing the cells to rely on OxPhos for energy. Indeed, growth in galactose increased the levels of most respiratory chain proteins in both control and all three mutant fibroblasts, and also in cells transduced with either the empty vector or TIMM50, resulting in higher respiration and ATP production, a phenomenon also reported in human primary myotubes (Aguer *et al*, 2011) and cancer cells (Rossignol *et al*, 2004). Despite this metabolic switch, no significant increase in the levels of ROS and related proteins was detected in galactose compared to glucose. By contrast, mitophagy markers p62 and LC3-II were both upregulated in galactose-grown cells without significant differences among cell lines. Thus, *TIMM50* mutant fibroblasts grown in galactose seem to have a relatively milder biochemical phenotype than those grown in glucose, which sharply contrasts with their strong reduction in cell growth and huge increase in apoptosis, an effect also seen in human primary skin fibroblasts from other patients with a variety of OxPhos deficiencies (Robinson *et al*, 1992). TIMM50 depletion has also proven to induce apoptosis in human breast cancer cell lines (Gao *et al*, 2016), whereas Tim50 loss of function in zebrafish resulted in massive apoptosis in the central nervous system associated with cytochrome c release (Guo *et al*, 2004). Moreover, mutant huntingtin specifically associates with the TIM23 complex and directly inhibits protein import, triggering neuronal cell death in mouse (Yano *et al*, 2014). A significant cell growth defect was also observed in *ISCU* mutant fibroblasts but in this case, it was not associated with increased apoptosis. These results highlight the importance of healthy mitochondrial function in tissues with high energy demand and show how the analysis of patient cells growing in galactose vs. glucose can provide an insight into the effect that these mutations may have in OxPhos-dependent tissues and organs.

# Materials and Methods

### Human subjects

Informed consent was obtained from all subjects included in the study, and the experiments conformed to the principles set out in the WMA Declaration of Helsinki and the Department of Health and Human Services Belmont Report.

### Whole exome sequencing analysis

Whole exome sequencing analysis was applied to 50 ng of genomic DNA from TIMM50 Pt. Nextera Rapid Exome Capture Kit (Illumina) was used for exome enrichment according to manufacturer's instructions. Exome sequencing was performed by Illumina MiSeq next-generation sequencing platform with 600-bp pair-end sequencing strategy. Sequences from the FASTAQ files were aligned to the human genome (hg19) by using the BWA aligner. Variants were called by using the GATK HaplotypeCaller and then filtered by using the Variant Quality Score Re-calibration according to the best practices of GATK-2.7 (https://www.broadinstitute.org/gatk/). Variants were annotated by using ANNOVAR. Coverage of the targeted regions was estimated using the GATK DepthOfCoverage. For paired-end reads to be included, they needed to have a mapping quality >20 and a base quality >10.

### Cell culture and transduction

Fibroblast cell lines were maintained in high-glucose medium (Gibco) supplemented with 10% FBS (Gibco), 1% penicillin-streptomycin, and 100 μg/ml uridine at 37°C in a humidified atmosphere of 5% $CO_2$. Primary skin fibroblasts were immortalized by lentiviral transduction of pLOX-Ttag-iresTK (Addgene #12246, Tronolab). For the generation of lentiviral particles, human 293T cells ($2.5 \times 10^6$) were plated 24 h before cotransfection with 10 μg of transfer vector (pLOX-Ttag-iresTK), 7.5 μg of second-generation packaging plasmid (pCMVdR8.74), and 3 μg of envelope plasmid (pMD2.VSVG; Naldini *et al*, 1996). FuGENE 6 Transfection Reagent (Roche) was used as transfectant reagent. Infectious particles were collected 24 and 48 h after transfection. Transduced fibroblasts were grown for at least six passages in order to make sure immortalized cells were selected. Changes in cell shape and doubling time were observed as part of the normal process of immortalization. Complementation experiments on immortalized fibroblasts used wild-type TIMM50S cloned into pWPXL-ires-Hygro[R] lentiviral expression vector, a modified version of pWPXLd (Addgene #12258, Tronolab) or the empty vector itself. Lentiviral particles were generated as described above, and transduced fibroblasts were selected for hygromycin resistance. When required, high-glucose medium was replaced by glucose-free medium (Gibco) and supplemented with 50 mM galactose (Sigma) for 5 days. Unless specified otherwise, all experiments were carried out on immortalized skin fibroblasts.

### Immunoblot analysis

Protein gel electrophoresis and blotting analyses were performed in fibroblasts (either naïve, immortalized cells and complemented) from patients and controls A and B. Samples from cell lysates containing 10 μg protein were separated by denaturing NuPAGE 4–12% Bis-Tris gels and transferred to nitrocellulose membrane. Immunodetection was carried out using antibodies against target proteins and GAPDH as loading control. In the case of cellular fractions, protein load was made equivalent to whole cell extract lane. A list of antibodies used in the present study is shown in Appendix Table S1.

## ROS measurement

Reactive oxygen species (ROS) was measured by monitoring intracellular oxidation of the cell-permeant probe 2′-7′ dichlorofluorescein diacetate (H$_2$DCFDA) in a spectrofluorimeter. Control and patient fibroblasts ($35 \times 10^3$ cells) grown in 96-well plates in glucose- or galactose-containing media were left untreated or treated with 250 μM H$_2$O$_2$ for 30 min before being incubated with 9 μM DCFDA in HBSS for 10–40 min, and fluorescence was measured at Ex 488 nm, Em 525 nm. Next, cells were lysed in 50 mM Tris–HCl 7.4/150 mM NaCl/1 mM EDTA/0.1% Triton and proteins were quantified. Fluorescence values were normalized to protein load.

## Annexin V assay

Quantification of apoptotic cells was carried out by measuring Annexin V and propidium iodide in a NucleoCounter NC-3000 Advanced Image Cytometer and following the manufacturer's recommendations.

## Oxygen consumption

Respiration in skin fibroblasts, $I_{O2}$ [pmols·s$^{-1}$·10$^{-6}$ cells], was calculated as the negative time derivate of oxygen concentration as measured by the OROBOROS Oxygraph-2k on one million cell/ml in a 2-ml chamber at 37°C. Basal respiration was measured without substrates and the proton leak state after the addition of oligomycin (50 nM). Oxygen consumption coupled to ATP production was calculated as the difference between basal respiration and proton leak. Maximal respiration was measured by stepwise 1.25 μM titration of CCCP and inhibition by 2 μM rotenone and 2.5 μM antimycin A for the final measurement of residual oxygen consumption. Spare capacity was calculated as the difference between maximal respiration and basal respiration.

## Membrane potential

Mitochondrial membrane potential was measured in untreated cells and after treatment with 1 μM FCCP for 5 min at 37°C as ratio of red to green JC-1 signal using a NucleoCounter NC-3000 Advanced Image Cytometer. In addition, confocal analysis based on the use of TMRM was also performed. Fibroblasts were washed with PBS twice and live-stained with 20 nM TMRM for 20 min. After washing three times with PBS, cells were imaged with a ZEISS ApoTome fluorescence microscope using a 40× objective within five minutes after staining.

## *In organello* import of radiolabeled proteins

[$^{35}$S]-methionine-labeled proteins were generated with the TNT Quick Coupled Transcription/Translation System (Promega) according to the manufacturer's instructions. Labeled proteins were incubated with isolated mitochondria prepared by differential centrifugation from control and patient fibroblasts grown in glucose (Reyes *et al*, 2011) and import were carried out at 37°C for 1–30 min. A negative control containing 1 μM FCCP was included for the longest time point. Mitochondria from TFAM import reactions were washed three times before being subjected to SDS–PAGE, while in AAC1 import experiments, mitochondria were trypsin

treated for 15 min at 37°C before the washes and SDS–PAGE. Gels were dried, exposed to storage phosphor screens (GE Healthcare), visualized on the Typhoon 9410 Variable Mode Imager (Amersham Biosciences), and quantified using ImageJ.

## qPCR

RNeasy mini kit (QIAGEN) and GoTaq 2-Step RT-qPCR System (Promega) were used for RNA purification from fibroblasts and cDNA retrotranscription, respectively, according to the manufacturers' protocols. The expression levels of *TIMM50* and the housekeeping genes *GAPDH* and *ACTB* were analyzed by GoTaq qPCR Master Mix (Promega), using the SYBR green chemistry. Primer sequences were as follows: *TIMM50* I (GGGACTGTGCACGAGGTT and CCTTGCGCCTTAGTGTCTC), *TIMM50* II (CTTTCAAACAGCGGCAAA and GGAGCCAAGGAAGAGGTTCT), *GAPDH* (CTCTGCTCCTCCT GTTCGAC and ACGACCAAATCCGTTGA), and *ACTB* (CCAACCGC GAGAAGATGA and CCAGAGGCGTACAGGGAT).

## Statistical analysis

Fibroblasts from a single patient with mutations in TIMM50 and two non-related healthy individuals along with two fibroblasts cell lines from patients with mutations in other genes affecting mitochondrial function were also analyzed as controls. All numerical data are expressed as mean ± standard deviation of the mean (SD). Student's unpaired two-tailed *t*-tests under the assumption of normal distribution and unequal variance were used for statistical analysis combining the data from both controls against the *TIMM50* patient unless specified otherwise. In complementation experiments, non-complemented, and complemented with either empty vector or TIMM50S, patient-derived cell lines were compared to the corresponding control B cell line that was randomly chosen for these experiments.

# Data availability

The datasets produced in this study are available in the following databases:
Whole exome sequencing: ClinVar SCV000788332 (https://www.ncbi.nlm.nih.gov/clinvar/variation/559480/) and SCV000788354 (https://www.ncbi.nlm.nih.gov/clinvar/variation/559479/).

Expanded View for this article is available online.

## Acknowledgments

Our work was supported by the Core Grant from the MRC (QQR 2015-2020); ERC Advanced Grant FP7-322424 and NRJ-Institut de France Grant (to M.Z.); by the Telethon Grant GGP15041 and the Pierfranco and Luisa Mariani Foundation (to DG); by the "Cell lines and DNA Bank of Genetic Movement Disorders and Mitochondrial Diseases" of the Telethon Network of Genetic Biobanks (grant GTB12001J).

## Author contributions

AR conceived the study and designed the experiments, immortalized the skin fibroblasts, performed all the experiments on immortalized fibroblasts, interpreted the results, and wrote the manuscript. LM performed segregation analysis and some preliminary experiments on naive mutant fibroblasts. AB took

**The paper explained**

**Problem**

Mitochondria are estimated to contain about 1,500 proteins out of which only 13 are synthesized inside the mitochondrial matrix. The remaining proteins need to be imported by specialized import machineries, the translocases of the mitochondrial outer and inner membrane: TOM and TIM complexes, respectively. TIMM50 is an essential component of the mitochondrial protein import machinery TIM23. In humans, only six diseases have been associated with mutations in components of the import machinery and only one previous case involved *TIMM50*.

**Result**

We found novel *TIMM50* mutations in an infant patient with rapidly progressive, severe encephalopathy, and we analyzed their pathogenicity in the patient-derived fibroblasts. Patient fibroblasts displayed a decrease in the protein levels of all components the TIM23 complex which resulted in impaired mitochondrial protein import through the TIM23 but not the TIM22 and a decreased in mitochondrial membrane potential. As a consequence, we observed lower levels of some key components of the mitochondrial respiratory chain which lead to decreased mitochondrial respiration and increased reactive oxygen species (ROS) production. Moreover, shifting energy production toward oxidative phosphorylation (OxPhos) by growing cells in galactose-containing media caused massive cell death. Complementation of patient fibroblasts with wild-type TIMM50 rescued or at least substantially improved the phenotype, confirming the observed mutations are causative of the disease. Furthermore, two additional patient-derived cell lines carrying mutations in *RNASEH1* or *ISCU* were analyzed and supported the idea that while some patterns are shared by all three cell lines, many of them were unique to *TIMM50* mutations.

**Impact**

The identification of novel *TIMM50* mutations resulting in impaired mitochondrial import and a clear mitochondrial dysfunction demonstrates for the first time the pathogenicity of mutations in this gene. Furthermore, the increased apoptosis observed in galactose media supports the critical role of this protein for cell viability in OxPhos-dependent condition which is an important observation that could help us understanding tissue-specific variations in mitochondrial diseases since some tissues and organs are more dependent on OxPhos than other for energy production.

care of the patient and wrote the clinical case. AJR provided bioinformatics management of the WES data. DG supervised preliminary biochemical, genetic, and cellular analyses on the case. MZ co-conceived the study and critically reviewed the manuscript.

## Conflict of interest

The authors declare that they have no conflict of interest.

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
