## [Review Process File · EMBO Molecular Medicine]

Mutations in *TIMM50* compromise cell survival in OxPhos-dependent metabolic conditions

Aurelio Reyes, Laura Melchionda, Alberto Burlina, Alan J. Robinson, Daniele Ghezzi and Massimo Zeviani.

Review timeline:

Submission date:	17 th November 2017
Editorial Decision:	7 th January 2018
Revision received:	19 th June 2018
Editorial Decision:	17 July 2018
Revision received:	2 nd August 2018
Accepted:	14 th August 2018

Editor: Céline Carret

Transaction Report:

1st Editorial Decision

7th January 2018

Thank you for the submission of your manuscript to EMBO Molecular Medicine. We have now heard back from the three referees whom we asked to evaluate your manuscript.

You will see below that they all find that the study is important and clinically relevant and should be revised for publication in EMBO Molecular Medicine. While we understand that the single patient data is unfortunately unavoidable, referees would greatly appreciate more (in vitro) data to increase the functional insights of the mutation and go towards verifying causation. Additional details, discussions and rewriting should also be done accordingly.

Upon our cross-commenting exercise, it became apparent that strengthening the part concerning galactose vs glucose effects is important and this could be perhaps achieved by searching for a similar phenomenon in other cellular models of mitochondrial diseases as you suggested in the discussion.

We would welcome the submission of a revised version within three months for further consideration and would like to encourage you to address all the criticisms raised as suggested to improve conclusiveness and clarity.

 REFEREE REPORTS.

Referee #1 (Comments on Novelty/Model System for Author):

Solid work

Referee #1 (Remarks for Author):

The manuscript by Reyes and colleagues describes the characterization of fibroblasts of a single

patient with mutations in the TIMM50 gene (identified by exam sequencing). Cells displayed lower mitochondrial membrane potential and impaired mitochondrial protein import through the TIM23 pathway. Cells also had decreased mitochondrial respiratory chain complexes I, II and IV. Growth of patient fibroblasts in galactose, a condition that shifts energy production metabolism towards oxidative phosphorylation (OxPhos), showed an improvement in respiratory function but also an increase in apoptosis.

The work is straightforward and well performed. Despite being a single patient, it is of broad interest as there are not many diseases caused by defective mitochondrial protein import. I have mostly minor comments.

- 1) I would like to have seen western blots for a larger number of TIMM50-dependent proteins.
- 2) The Import data with AAC1 is not very clear looking at the figure. Not having the mature form, what does the increase intensity at 10 minutes mean??
- 3) I found the Introduction too long. I suggest streamlining it.
- 4) Page 9 misspelled "clinical discussion ofn all the"

Referee #2 (Remarks for Author):

The manuscript by Reyes and co-authors reports a study on fibroblasts of a patient with TIMM50 mutations. The manuscript is well written and the data are clearly presented and discussed. Nevertheless, I have some comments and questions.

TIMM50 mutations have been previously reported in patients with intellectual disability and seizures (Shahrour et al, 2017). The clinical presentation of the patient depicted in the present manuscript is relatively similar but none of the clinical signs in any patient is specific as they have been reported in several other patients with other gene mutations. No functional validation of the TIMM50 has been done in Shahrour paper hampering to demonstrate that the mutations are indeed disease causing. Therefore, it is difficult to judge if the identified mutations are actually causative, although decreased TIMM50 protein amount is demonstrated in patient fibroblasts. The authors should present additional functional data to convince that TIMM50 is the disease gene.

Overall, this manuscript reports a detailed characterization of TIMM50 mutant fibroblasts (steady state levels of other components of the import machinery, efficiency of mitochondrial import, membrane potential, OXPHOS protein level and respiration, ROS production, mitophagy, cell size, cell growth and apoptosis). Obviously, the most interesting data are those showing abnormal import of mitochondrial protein that could be related to the function of TIMM50. However, it is unclear if all other features are specific to TIMM50 defect. Indeed, several articles report such abnormal phenotypes associated with several other mitochondrial defect. In order to demonstrate the specificity of these phenotypes this study should include one or more other patient cell lines with mutations affecting other mitochondrial functions. Otherwise, these data are a little bit useless. Moreover, the discussion of all these data is relatively long and should be shortened, especially if the data are not specific to TIMM50 defect.

Introduction. The authors listed the mutations in genes encoding components of the mitochondrial protein import machinery. Abnormal mitochondrial protein import is very rare in human genetics and it should be interesting to also cite the MAGMAS/PAM16 mutations even if they result in a non-mitochondrial disease.

Figure 1B. Protein alignment shows that G190 is relatively conserved among species reinforcing its pathogenicity. However, the protein alignment around S112 is useless as the c.335C>A mutation induces a premature stop codon.

Figure 2D. TIMM50 protein appears as a double band in this western blot experiment. Could the authors comment this point?

The authors questioned if TIMM50 mutations affect the protein level of other import components by measuring the steady state levels of various proteins in cells grown in glucose and galactose media

(Fig. 3). TOMM40 gave strange results as its level was very decreased in the two control cell lines and much lower than in TIMM50 fibroblasts. Is that due to an experimental problem or not? This data should be checked and commented if confirmed.

Figure 6B. Controls A and B displayed significant difference in SOD2 amount when grown in glucose media. The authors should comment this point. When grown in galactose medium TIMM50 fibroblasts showed increased SOD2 levels suggesting increased oxidative stress. This stress should alter Fe-S clusters and therefore decrease ACO2 protein which is an Fe-S protein. However, ACO2 is increased. Do the authors have any explanation?

No or few information about the number of technical and biological replicates for the experiment is reported.

Referee #3 (Comments on Novelty/Model System for Author):

The study is performed on fibroblasts of one patient. To build more general conclusions broader spectrum of pathology models is desired.

Referee #3 (Remarks for Author):

The manuscript described a new mutation with pathogenic consequences in TIMM50, an important component of the TIM23 translocase responsible for import of mitochondrial proteins. This is a detailed study of fibroblasts derived from patients. First, the authors find the levels of the core components of the TIM23 complex affected in the patient-derived cell lines, and this correlates well with defective presequence-driven protein import. They investigate bioenergetic properties and find that the patient cells are suboptimal in many bioenergetics aspects, including the inner membrane electrochemical potential, which was greatly reduced. In agreement, poor bioenergetics performance of patients fibroblasts is likely caused by the reduced abundance of some components of the respiratory complexes I, II and IV.

Interestingly, when patient fibroblasts have been grown on galactose, the conditions promoting oxidative phosphorylation, they are able to partially revert the described defects. However, a massive cell death is observed.

Overall, this is an interesting and important story with well-done experiments and convincing conclusions (in majority). This study is performed on one cell line derived from one patient. Also, the advance in understanding of mechanisms of pathology is quite limited and not surprising based on the current knowledge.

The observation on differences between glucose and galactose is really interesting and bears high potential but it is not developed further. Patient fibroblasts grown on glucose apparently can tolerate a relatively big decrease in the import components resulting in import efficiency drop. It can be hypothesized that under galactose promoting respiration this import defect is not tolerated and the large amount of cells dies. Those, which survive, have adopted to the defective import due to a very strong selection to overcome import defects. Is such phenomenon common for other mitochondria- or mitochondrial import-defective cells? What is the nature of such change that allows to survive with the defect? What is better: to have a decreased biogenesis of mitochondria/mitochondrial proteins or to face massive cell death in terms of maintain relative health vs pathology on the cell and organismal level?

I think that authors touch a very important problem, prerequisite to understanding tissue specificity and progression of mitochondrial diseases.

Minor points:

- In the introduction, the authors should mention the pathology related to ALR,
- TIMM17B should also be assessed as a more common isoform of TIMM17,
- The imports in organello experiments lack controls (membrane potential dissipation). Whereas the results of TFAM import are clear due to a proteolytic change in mass observed upon transfer to mitochondria, the import assay for AAC1 is not controlled for its specificity.
- Measurement of ROS productions: the strategy with hydrogen peroxide pretreatment is not clear.

REFeree REPORTS.

Referee #1 (Comments on Novelty/Model System for Author):

Solid work

Referee #1 (Remarks for Author):

The manuscript by Reyes and colleagues describes the characterization of fibroblasts of a single patient with mutations in the TIMM50 gene (identified by exam sequencing). Cells displayed lower mitochondrial membrane potential and impaired mitochondrial protein import through the TIM23 pathway. Cells also had decreased mitochondrial respiratory chain complexes I, II and IV. Growth of patient fibroblasts in galactose, a condition that shifts energy production metabolism towards oxidative phosphorylation (OxPhos), showed an improvement in respiratory function but also an increase in apoptosis.

The work is straightforward and well performed. Despite being a single patient, it is of broad interest as there are not many diseases caused by defective mitochondrial protein import. I have mostly minor comments.

1) I would like to have seen western blots for a larger number of TIMM50-dependent proteins.

We added several more Western-blot in Figure 2A, B such as TIMM17B, DNAJC19, Mortalin, GrpE and Magmas

2) The Import data with AAC1 is not very clear looking at the figure. Not having the mature form, what does the increase intensity at 10 minutes mean??

AAC1 import experiments have been redone with different timepoints and no decrease after 10 min of incubation during import is now detected.

3) I found the Introduction too long. I suggest streamlining it.

We have deleted some paragraphs. On the other hand, we think that a thorough description of the translocation machinery of mitochondria is necessary to allow a naïve but interested reader to follow the experimental framework and results.

4) Page 9 misspelled "clinical discussion ofn all the"

Thank you

Referee #2 (Remarks for Author):

The manuscript by Reyes and co-authors reports a study on fibroblasts of a patient with TIMM50 mutations. The manuscript is well written and the data are clearly presented and discussed. Nevertheless, I have some comments and questions.

TIMM50 mutations have been previously reported in patients with intellectual disability and seizures (Shahrour et al, 2017). The clinical presentation of the patient depicted in the present manuscript is relatively similar but none of the clinical signs in any patient is specific as they have been reported in several other patients with other gene mutations. No functional validation of the TIMM50 has been done in Shahrour paper hampering to demonstrate that the mutations are indeed disease causing. Therefore, it is difficult to judge if the identified mutations are actually causative, although decreased TIMM50 protein amount is demonstrated in patient fibroblasts. The authors should present additional functional data to convince that TIMM50 is the disease gene.

We clearly demonstrate that not only TIMM50 is decreased but also other components of the TIM23 machinery and whilst import of a TIMM22-dependent protein, AAC1, is not affected, the import of

a TIM23-dependent protein, TFAM, is virtually abolished in the presence of mutant TIMM50. In addition, all the defects in bioenergetics and related pathways found in TIMM50 mutant cells are rescued by the expression of transduced wild-type TIMM50.

Overall, this manuscript reports a detailed characterization of TIMM50 mutant fibroblasts (steady state levels of other components of the import machinery, efficiency of mitochondrial import, membrane potential, OXPHOS protein level and respiration, ROS production, mitophagy, cell size, cell growth and apoptosis). Obviously, the most interesting data are those showing abnormal import of mitochondrial protein that could be related to the function of TIMM50. However, it is unclear if all other features are specific to TIMM50 defect. Indeed, several articles report such abnormal phenotypes associated with several other mitochondrial defect. In order to demonstrate the specificity of these phenotypes this study should include one or more other patient cell lines with mutations affecting other mitochondrial functions. Otherwise, these data are a little bit useless. Moreover, the discussion of all these data is relatively long and should be shortened, especially if the data are not specific to TIMM50 defect.

The complementation experiments demonstrate that the defects in cells are clearly due to defective TIMM50 and can be rescued by the expression of the WT protein. We have now included two additional disease cell line as controls from patients with mutations in *RNASEH1* and *ISCU*, respectively, providing additional evidence that several features of the TIMM50 cell line are specific.

Introduction. The authors listed the mutations in genes encoding components of the mitochondrial protein import machinery. Abnormal mitochondrial protein import is very rare in human genetics and it should be interesting to also cite the MAGMAS/PAM16 mutations even if they result in a non-mitochondrial disease.

We have added the requested citations.

Figure 1B. Protein alignment shows that G190 is relatively conserved among species reinforcing its pathogenicity. However, the protein alignment around S112 is useless as the c.335C>A mutation induces a premature stop codon.

For the sake of clarity and completeness we have preferred to maintain both mutations in the figure.

Figure 2D. TIMM50 protein appears as a double band in this western blot experiment. Could the authors comment this point?

The doublet in TIMM50 is a constant in all fibroblast cell lines analysed in the present study. Actually, both bands were decreased in the patient cell line and increased in the transduced lines. Further studies will be required in order to get more information about the nature of these two bands.

The authors questioned if TIMM50 mutations affect the protein level of other import components by measuring the steady state levels of various proteins in cells grown in glucose and galactose media (Fig. 3). TOMM40 gave strange results as its level was very decreased in the two control cell lines and much lower than in TIMM50 fibroblasts. Is that due to an experimental problem or not? This data should be checked and commented if confirmed.

We have redone the Western Blot and the result shows a much milder difference in TOMM40 amount which is significantly different from controls.

Figure 6B. Controls A and B displayed significant difference in SOD2 amount when grown in glucose media. The authors should comment this point. When grown in galactose medium TIMM50 fibroblasts showed increased SOD2 levels suggesting increased oxidative stress. This stress should alter Fe-S clusters and therefore decrease ACO2 protein which is an Fe-S protein. However, ACO2 is increased. Do the authors have any explanation?

We have redone the WB and ACO2 is slightly increased in TIMM50 mutants. As expected, it is decreased in ISCU mutant cell line which implies a defect in the biosynthesis of Fe-S centres. We attribute the increase of ACO2 to increased ROS production, as it also occurs for SOD2, as some papers report an anti-ROS effect of ACO2.

No or few information about the number of technical and biological replicates for the experiment is reported.

Figure legends report now a full description of the number of replicates and their biological significance.

Referee #3 (Comments on Novelty/Model System for Author):

The study is performed on fibroblasts of one patient. To build more general conclusions broader spectrum of pathology models is desired.

Referee #3 (Remarks for Author):

The manuscript described a new mutation with pathogenic consequences in TIMM50, an important component of the TIM23 translocase responsible for import of mitochondrial proteins. This is a detailed study of fibroblasts derived from patients. First, the authors find the levels of the core components of the TIM23 complex affected in the patient-derived cell lines, and this correlates well with defective presequence-driven protein import. They investigate bioenergetic properties and find that the patient cells are suboptimal in many bioenergetics aspects, including the inner membrane electrochemical potential, which was greatly reduced. In agreement, poor bioenergetics performance of patients fibroblasts is likely caused by the reduced abundance of some components of the respiratory complexes I, II and IV.

Interestingly, when patient fibroblasts have been grown on galactose, the conditions promoting oxidative phosphorylation, they are able to partially revert the described defects. However, a massive cell death is observed.

Overall, this is an interesting and important story with well-done experiments and convincing conclusions (in majority). This study is performed on one cell line derived from one patient. Also, the advance in understanding of mechanisms of pathology is quite limited and not surprising based on the current knowledge.

The observation on differences between glucose and galactose is really interesting and bears high potential but it is not developed further. Patient fibroblasts grown on glucose apparently can tolerate a relatively big decrease in the import components resulting in import efficiency drop. It can be hypothesized that under galactose promoting respiration this import defect is not tolerated and the large amount of cells dies. Those, which survive, have adopted to the defective import due to a very strong selection to overcome import defects. Is such phenomenon common for other mitochondria- or mitochondrial import-defective cells? What is the nature of such change that allows to survive with the defect? What is better: to have a decreased biogenesis of mitochondria/mitochondrial proteins or to face massive cell death in terms of maintain relative health vs pathology on the cell and organismal level?

I think that authors touch a very important problem, prerequisite to understanding tissue specificity and progression of mitochondrial diseases.

Thank you for your comments. We do believe that apoptosis is operating as a selective pathway when mutant cells are grown in galactose, therefore forced to rely on OxPhos. This can determine the survival of the most proficient cells and therefore the paradoxical effect of an apparent amelioration of the bioenergetics in mitochondria of the surviving cells. The mechanistic, biological effects leading to this interesting phenomenon implies probably a specific, complex analysis, e.g. transcriptomic, metabolomic, etc. which is clearly beyond the scope of the present paper. Moreover, this galactose related apoptosis has not been observed in *RNASEH1* nor in *ISCU* mutant fibroblasts.

Minor points:

- In the introduction, the authors should mention the pathology related to ALR,

The pathology related to ALR is now included in the introduction.

- TIMM17B should also be assessed as a more common isoform of TIMM17,

TIMM17B has now been assessed in controls and patient cell lines and also in complementation experiments (Figures 2A, 2B, Appendix Figure S8A)

- The imports in organello experiments lack controls (membrane potential dissipation). Whereas the results of TFAM import are clear due to a proteolytic change in mass observed upon transfer to mitochondria, the import assay for AAC1 is not controlled for its specificity.

We have now used FCCP as a negative control at the longest time point and the results included in Figure 3.

- Measurement of ROS productions: the strategy with hydrogen peroxide pretreatment is not clear.

The pretreatment with H₂O₂ increases the levels of ROS in cells and enhances differences between mutant and control cells that were not so obvious in naïve conditions. This is now explained in the text and as it is a standard procedure in ROS measurement.

2nd Editorial Decision

17 July 2018

Thank you for the submission of your revised manuscript to EMBO Molecular Medicine. We have now received the enclosed reports from the referees that were asked to re-assess it. As you will see the reviewers are now globally supportive and I am pleased to inform you that we will be able to accept your manuscript pending final editorial amendments [not listed].

REFEREE REPORTS.

Referee #2 (Comments on Novelty/Model System for Author):

The manuscript reports TIMM50 mutations in a new patients. The data are convincing.

Referee #2 (Remarks for Author):

The authors correctly responded to my comments and questions. They did adequate new experiments.

Referee #3 (Remarks for Author):

The authors adequately addressed the criticism. This is an interesting and valuable work.

Corresponding Author Name: Aurelio Reyes and Massimo Zeviani

Manuscript Number: EMM-2017-08698